# SARS-CoV-2: Searching for the Missing Variants

**DOI:** 10.3390/v14112364

**Published:** 2022-10-26

**Authors:** Emilia Caputo, Luigi Mandrich

**Affiliations:** 1Institute of Genetics and Biophysics-IGB-CNR, “A. Buzzati-Traverso”, Via Pietro Castellino 111, 80131 Naples, Italy; 2Research Institute on Terrestrial Ecosystems-IRET-CNR, Via Pietro Castellino 111, 80131 Naples, Italy

**Keywords:** SARS-CoV-2, SARS-CoV-2 variants, phylogenetic analysis

## Abstract

Structural and phylogenetic analysis of the spike glycoprotein highlighted that the last variants, annotated as omicron, have about 30 mutations compared to the initial version reported in China, while the delta variant, supposed to be the omicron ancestor, shows only 7 mutations. Moreover, the five omicron variants were isolated between November 2021 and January 2022, and the last variant BA.2.75, unofficially named centaurus, was isolated in May 2022. It appears that, since the isolation of the delta variant (October 2020) to the omicron BA.1 (November 2021), there was an interval of one year, whereas the five omicron variants were isolated in three months, and after a successive four months period, the BA.2.75 variant was isolated. So, what is the temporal and phylogenetic correlation among all these variants? The analysis of common mutations among delta and the omicron variants revealed: (i) a phylogenetic correlation among these variants; (ii) the existence of BA.1 and BA.2 omicron variants a few months before being isolated; (iii) at least three possible intermediate variants during the evolution of omicron; (iv) the evolution of the BA.2.12.1, BA.4 and BA.5 variants from omicron BA.2; (v) the centaurus variant evolution from omicron BA.2.12.1.

## 1. Introduction

The last three years have been characterized by a worldwide pandemic due to a new type of coronavirus, called SARS-CoV-2, and reported for the first time in China in December 2019 [1]. In September 2022, WHO confirmed about 610 million cases and 6.5 million deaths by SARS-CoV-2 (https://www.who.int/publications/m/item/weekly-epidemiological-update-on-COVID-19, 28 September 2022).

SARS-CoV-2 is derived from the *Coronaviridae* viruses group [2]. It has a genome constituted of a positive sense single-stranded RNA of about 30 kb length [3], its mechanisms of infection and replication have been elucidated and a critical rule has been assigned to the membrane spike glycoprotein (S) [3,4]. Since spike protein is the most abundant and characteristic protein of SARS-CoV-2, it has been used as an antigen for the vaccine’s production against the virus [4]. Furthermore, the mutations isolated on the spike protein have been used to classify and monitor the SARS-CoV-2 variants [5,6]. In fact, about 30 SARS-CoV-2 variants have been identified and classified as either Variants of Concern (VOCs) or Variants of Interest (VOIs). The VOCs show a greater virulence, transmissibility and severity of the symptoms compared to VOIs, as well as a reduced effectiveness of the vaccines [5,7]. The VOIs, instead, are mainly characterized by alterations into the receptor binding affinity [7]. According to this classification, the variants alpha, beta, gamma, delta and omicron are defined as VOCs, whereas lambda and mu are defined as VOIs [7]. Furthermore, each VOC is able to accumulate significant mutations, resulting in a rapid replacement of previous variants [7].

The SARS-CoV-2 variants have accumulated mutations, affecting the infection and diffusion of viral mechanisms, in particular, an increase in diffusion and milder symptoms have been observed [8,9]. It is important to note that SARS-CoV-2 variants are also characterized by other genomic mutations that are not present on spike protein; this is the case for the omicron BA.4 and BA.5 variants that differ from the initial version of the virus for about 50 mutations: 30 of these are on spike protein, and both variants have the same mutations, whereas they differ for other genomic mutations, which has led to having two variants with different genotypic and phenotypic characteristics [10].

Here, we analyzed at structural level the six omicron variants that have indications about their origin and phylogenesis, because they have been identified in a restricted interval time, from November 2021 to January 2022 [10], but they preset a high number of mutations in respect to the last isolated mu variant, in January 2021 [11,12].

## 2. Materials and Methods

The sequences of SARS-CoV-2 variants delta, omicron BA.1, BA.2, BA.2.12.1, BA.4, BA.5 and BA.2.75 were from the “expasy viralzone” web site (https://viralzone.expasy.org/9556, 17 September 2022), and were used to make the multiple sequence alignment to generate a phylogenetic three (Multiple Sequence Alignment by Clustal Omega program at https://www.ebi.ac.uk/Tools/msa/clustalo/; 17 September 2022). 

## 3. Results

In a recent study, the SARS-CoV-2 variants were analyzed, at structural level, in order to generate a phylogenetic three, indicating a common ancestor between the delta and the five SAR-CoV-2 omicron variants [12], although the spike protein mutations on the delta variant were 7 compared to the more than 30 ones identified on the omicrons [12]. Moreover, the mu variant, reported in January 2021, was the last variant isolated before omicron BA.1 in November 2021. Successively, until January 2022, the other five omicron variants were reported, suggesting either a delay in the identification of new variants or the possibility that they have not been identified at all.

In some cases, there have been countries with high numbers of infections where the virus has evolved rapidly [13] and perhaps for politic, economic and/or technological reasons, few sequences of the viral genome have been performed; in other cases, especially in the countries where the vaccination against SARS-CoV-2 has reached high levels of coverage in the population, it was thought that it had defeated the virus and therefore the monitoring of any new variants was reduced, instead the new variants accumulating many mutations escape the antibody response, following by natural infection or vaccination [14,15].

Here, we have analyzed the sequence alignment and the common mutations among the omicron variants to reconstruct their evolutionary lineage and to identify any intermediate variants that have never been reported.

### 3.1. The Omicrons Origin and Evolution

The omicron variants are characterized by a high number of mutations compared to the initial version of the virus. Since the SARS-CoV-2 variants, isolated before omicron, are characterized by a low number of mutations, the omicron origin seems to be uncertain showing common mutations described in other VOCs, such as alpha, beta, gamma and delta variants [10,16], but from an evolutionary point of view, the delta variant seems to be closer to omicron than the others [12]. Thus, we cannot exclude that the omicron origin could be derived by events of genomic recombination in two VOCs, contemporarily infecting patients. Furthermore, an antigenic shift has been observed for the omicron variant that is a step change from the viral antigenicity, leading to viral escape from vaccine-acquired immunity or infection from previous variants, which is consistent with the observed increased transmissibility [17].

Starting from this information, we generated a new multiple sequences alignment among the wild type version of the spike protein and the variants delta, omicron BA.1, BA.2, BA.2.12.1, BA.4, BA.5 and the last BA.2.75, unofficially indicated as centaurus (see Appendix A). We identified all the spike common mutations among the variants and those present only in one or more variants, as schematically represented in Figure 1.

Despite the omicron variants evolved from delta, they show only two common mutations: T478K and D614G. The position 681 is mutated in delta and omicron variants, but in delta, the substitution was P681R whereas in omicron, it was P681H. The other mutations present only in delta are T19R, the deletion 156-158 with substitution in glycine, L452R and D950N.

Interestingly, the alignment revealed the common mutations in all the omicron variants, which are S373P, S375F, K417N, N440K, S477N, T478K, E484A, N501Y, Y505H, D614G, H655Y, N679K, P681H, N764K, D796Y, Q954H and N969K (Figure 1, Appendix A).

Sequence analysis suggests that omicron BA.1 and BA.2 are evolved independently from a common ancestor, derived from delta and also having the mutations T478K and D614G (Figure 1, Appendix A); in fact, omicron BA.1 presents nine unique and characteristic mutations, which are A67V, T95I, deletion 142-145 with substitution in aspartic, deletion 211-212, G446S, G496S, T547K, N856K and L981F, whereas omicron BA.2 shows the deletion 24-27 as a specific mutation with a substitution in serine (Figure 1, Appendix A). 

Omicron BA.2.12.1 evolved from BA.2 because it has two other specific mutations in common: S375F and Q493R, and the unique and specific mutation S704L (Figure 1, Appendix A). From omicron BA.2.12.1, BA.4/BA.5 and BA.2.75 are separately evolved; the spike protein of the variants BA.4 and BA.5 shows the same mutations and both differ from the spike protein of BA.2.12.1 because it did not have the S704L and Q493R mutations, while presenting the deletion 69-70 in common with omicron BA.1 (Figure 1, Appendix A).

The last isolated variant, BA.2.75, shows the 17 common mutations with the other omicrons and the unique and specific mutations K147E, W152R, F157L, I210V, G257S, G446N and N460K, indicating that BA.2.75 evolved from BA.2.12.1 (Figure 1, Appendix A).

### 3.2. The Omicron BA.4 and BA.5

As reported, BA.4 and BA.5 variants have the same mutations on the spike proteins and they are similar to the BA.2.12.1 one, but they are classified as two different variants [10]. This is possible because they differ in about 50 mutations on the whole genome from the initial version of SARS-CoV-2, and 30 of them are on spike (Figure 1, Appendix A), while about 20 are in other part of the genome.

In particular, they show a similar pattern at 5′ genome region, corresponding to the genes encoding ORF1ab and the envelope protein E. In addition, while BA.4 shows specific mutations at ORF7b, N protein and nonstructural protein 1 (NSP1), both associated to the genomic RNA, BA.5 presents specific differences at the 3′ genome region, in correspondence with ORF6 and membrane protein M encoding genes [10].

### 3.3. The Omicron BA.2.275

The last SARS-CoV-2 variant identified is omicron BA.2.75, unofficially known as centaurus. It was isolated in May 2022 in India [18], and it is characterized by 34 mutations on the spike protein, 17 of them are in common with the other omicron variants. Further, it shows the reversion of R493Q compared to the ancestral variant, and seven unique and specific mutations: K147E, W152R, F157L, I210V, G257S, G446N and N460K (Figure 1, Appendix A) [18,19], which may be related to immune escape and resistance to antibody therapies, indicating a typical antigenic shift [20]. There are indications that the BA.2.75 variant was isolated for the first time in India in January 2022, and spread outside India only in May 2022. The symptoms associated with this variant are fever, cough, sputum, diarrhoea and fatigue [20], similar to those of a seasonal flu, as previously predicted [10].

## 4. Discussion

The analysis of the alignment of the wild type version, delta and the omicron variants highlight important information about the origin and the evolution of these variants. The timing of their evolution is different from their isolation date. In fact, they are not evolved in sequence from BA.1 up to BA.2.75 but through a number of intermediate variants, which have never been isolated. In particular, by analyzing all the mutations of these variants, we obtained a more correct evolutionary lineage that includes at least three intermediates, identified through the common mutations among them (Figure 2). We supposed that there was a joint starting from the wild type version of spike protein, represented by Intermediate 1 (Figure 2), where the evolutionary branches of delta and omicron variants have separated, and this missing intermediate only has the two common mutations among them, which are T478K and D614G. From Intermediate 1 evolved Intermediate 2, which shows the 17 mutations present in all the omicron variants. In Table 1, the mutations, that we assigned to the missing SARS-CoV-2 variants in the omicron lineage are listed, indicated as Intermediate 1, 2 and 3 (Table 1).

The variants BA.1 and BA.2 evolved separately from Intermediate 2 (Figure 2); BA.1 harbors 35 mutations in its spike protein in respect to the initial version isolated in SARS-CoV-2, 10 of them are specific and not present in the other omicron. Specifically, they are A67V, DEL69-70, T95I, DEL142-145D, DEL211-212, G446S, G496S, T547K, N856K and L981F (Figure 1); whereas BA.2 lacks 13 mutations present in BA.1 but shows 8 unique mutations not found in BA.1 that are T19I, DEL24-27S, G142D, V213G, T376A, D405N, R408S and Q498R [21]. Based on the presence of a high number of different mutations between them, it is reasonable to assume that their evolution process began and ended a few months before their isolation.

The variant BA.2.12.1 evolved from BA.2. In fact, beyond the common mutations in spike protein, BA.2.12.1 has also the substitutions T19I, L452Q and S704L [22] (Figure 2). 

Following the scheme of Figure 2, BA.4, BA.5 and BA.2.75 evolved from the BA.2.12.1 variant. The SARS-CoV-2 variant classification also takes into account other mutations present in the genome that change the characteristics of the virus. In fact, the spike proteins of BA.4 and BA.5 are identical but they differ for mutations in E and M genes [10]. Considering that the deletion 69-70 is present also in BA.1 (Figure 1) and in alpha variant, it has been suggested that BA.4 and BA.5 diverged via recombinant event with other variants [22]. Another study suggested that BA.4 and BA.5 evolved by a recombination with mouse SARS-CoV-2 virus, and in this case, mouse would have been the host for the evolution [23]. Regarding our analysis, it suggests that BA.4 and BA.5 evolved from BA.2.12.1 (Figure 2). In this case, we identified a third intermediate having all the mutations only on spike protein present in BA.4 and BA.5, and not on the other regions of the viral genome. Then, subsequent mutations on the genome led to the two variants BA.4 and BA.5; in Table 1 are listed the mutations characterizing the Intermediate 3.

The last omicron variant to be isolated was BA.2.75; it evolved from BA.2.12.1 (Figure 2). Sequence analysis highlighted that BA.2.75 compared to BA.2.12.1 shows seven new specific mutations: K147E, W152R, F157L, I210V, G275S, G446N and N460K, and reverted to the mutation Q493R (Figure 1) [18]. Epidemiological data reveal that omicron BA.2.75 is present in more than 30 countries worldwide and it does not appear to be dangerous, keeping the hospitalization of patients unchanged [24]. Probably, it is necessary to study in more detail the characteristics of this new variant and to use the new vaccines designed against both the BA.4/5 and BA.2.75 variants, to reduce the risk of a new critical healthcare wave of infection.

## 5. Conclusions

During the pandemic, about 30 variants were isolated, but in many cases, it was not possible to follow their evolution in time. After three years of the pandemic, we observed a new evolutionary lineage represented by the omicron variants, which have completely replaced the pre-existing variants. The timeline of the omicron evolution variants is different from the date of their isolation, in fact, a retroactive analysis revealed that omicron BA.1 was present in Europe at least 10 days before its isolation in South Africa [25]. Moreover, five different omicron variants were reported in three months, which differ among them from 8 to 18 mutations (Figure 1), suggesting that many intermediate variants were lost.

This study referred to a small group of evolutionarily correlated variants and strongly supported the existence of at least three important intermediates, which never have been isolated. To date, if we consider that about 30 SARS-CoV-2 variants have been isolated, we can state that there are many other missing variants.

## Figures and Tables

**Figure 1 viruses-14-02364-f001:**
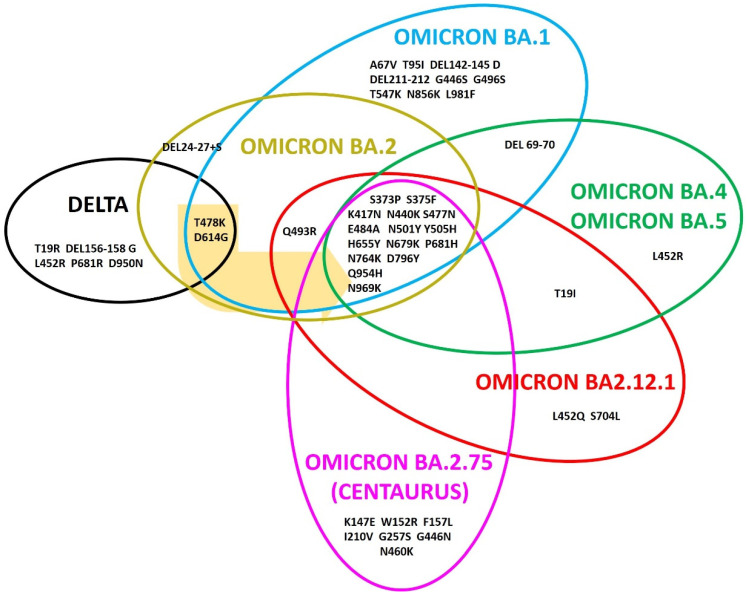
Schematic representation of the omicron variants evolution starting from delta. The common mutations are reported at the overlap among the variants. The mutations present in all the omicron variants are S373P, K417N, N440K, S477N, T478K, E484A, N501Y, Y505H, D614G, H655Y, N679K, P681H, N764K, D796Y, Q954H and N969K, and T478K and D614G derived from delta, indicated by the yellow arrow. The unique mutations of each variant are reported outside the overlap among the variants.

**Figure 2 viruses-14-02364-f002:**
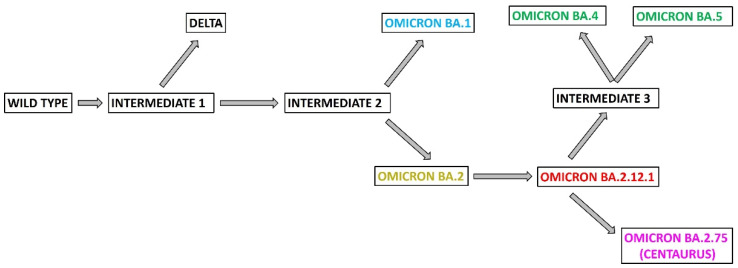
Phylogenetic lineage of the omicron variants starting from wild type version and the delta variant. Intermediates 1, 2 and 3 are three missing variants. Intermediate 1 has common mutations between delta and Intermediate 2. Intermediate 2 has the common mutations of all the omicron variants. Intermediate 3 has the common mutation between omicron BA.4 and BA.5.

**Table 1 viruses-14-02364-t001:** List of mutations that are supposed to be present in the three missing intermediates related to the omicron lineage of evolution.

Intermediate 1	Intermediate 2	Intermediate 3
T478K D614G	S373P S375F K417N N440K S477N T478K E484A N501Y Y505H D614G H655Y N679K P681H N764K D796Y Q954H N969K	T19I DEL24-27S DEL69-70 G142D V213G G339D S371F S373P S375F T376A D405N R408S K417N N440K L452R S477N T478K E484A F486V Q498R N501Y Y505H D614G H655Y N679K P681H N764K
		D796Y Q954H N969K

## Data Availability

Not applicable.

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
