# Peer review of "SARS-CoV-2: Searching for the Missing Variants"

_viruses, 2022, doi:10.3390/v14112364_

Round 1

Reviewer 1 Report

Dear authors,

Congratulations for your work.

 Here are my comments:

 Introduction:

Line 29 – Unable to open https://www.who.int/emergencies/diseases/novel- 29 coronavirus-2019.  The internet site reports that the item you requested could not be found.

Material and method:

It would be good to develop the material and method section. Why did you choose these methods?

 Results

-

 Discussion

-

 Conclusion

Does the information in the Conclusions chapter meet the intended purpose?

Author Response

Line 29 – Unable to open https://www.who.int/emergencies/diseases/novel- 29 coronavirus-2019.  The internet site reports that the item you requested could not be found.

We apologise for the error, and we inserted the correct link:

https://www.who.int/publications/m/item/weekly-epidemiological-update-on-covid-19---28-september-2022

Material and method:

It would be good to develop the material and method section. Why did you choose these methods?

In the section Materials and Methods, we reported the in silico analysis of the SARS-CoV-2 omicron variants to verify their structural relationship; our results about the identification of the putative intermediates in the omicron lineage has been reported in the Results section. We used the Clustal Omega program for the alignment, because we think that it is one of the best program for protein alignment, and also because this paper is a further progress to our previous paper: Caputo E., Mandrich L. (2022) Structural and Phylogenetic Analysis of SARS-CoV-2 Spike Glycoprotein from the Most Widespread Variants. Life (Bassel). 12(8):1245. doi: 10.3390/life12081245, where we made an in silico analysis by Clustal Omega program to create a phylogenetic tree of the most widespread SARS-CoV-2 variants.

Conclusion

Does the information in the Conclusions chapter meet the intended purpose?

Yes, we have achieved our goal. We proposed a method to study the evolution of the SARS-CoV-2 and identify putative intermediate variants that were lost.

Reviewer 2 Report

The authors traced the evolutionary lineage of omicron and discussed the omicron subvariants, with an aim to find traceable information regarding how the omicron variant varies. The study is interesting and sounds well. A few points are suggested for a change in writing:

1. Authors need to briefly discuss the evolution of VOCs, before putting a focus on the omicron. Relations between previous VOC and current dominating omicron may be delineated. 

2. Vital genetic changes in omicron subvariants should be linked to the important virological alterations, as it should be more discussed in the text. 

3. English needs improvement. The format of the text, such as in the Abstract, should be carefully adjusted. 

Author Response

  1. Authors need to briefly discuss the evolution of VOCs, before putting a focus on the omicron. Relations between previous VOC and current dominating omicron may be delineated.

To better explain this point, we inserted at Page 1 line 41 the sentence: “Actually, about thirty SARS-CoV-2 variants have been identified and classified as Variant of Concern (VOC) and Variant of Interest (VOI). The VOCs show a greater virulence, transmissibility, and severity of the symptoms compared to VOIs, as well as a reduced effectiveness of the vaccines [5, 7]. The VOIs, instead, are mainly characterized by alterations into the receptor binding affinity [7]. According to this classification, the variants alpha, beta, gamma, delta and omicron are defined as VOCs, whereas, lambda and mu as VOIs [7]. Furthermore, each VOC is able to accumulate significant mutations, resulting in a rapid replacement of previous variants [7].”

  1. Vital genetic changes in omicron subvariants should be linked to the important virological alterations, as it should be more discussed in the text.

To better explain this point, we inserted at page 2 line 96 the phrase: Thus, we cannot exclude that the omicron origin could be derived by events of genomic recombination in two VOCs, infecting patient contemporarily. Furthermore, for the omicron variant it has been observed an antigenic shift that is a step change of the viral antigenicity, leading to viral escape from vaccine-acquired immunity or infection from previous variants, which is consistent with the observed increased transmissibility [17].”

  1. English needs improvement. The format of the text, such as in the Abstract, should be carefully adjusted.

We have corrected the English style of the paper.